# ASK YOUR HUMANS: USING HUMAN INSTRUCTIONS TO IMPROVE GENERALIZATION IN REINFORCEMENT LEARNING

**Valerie Chen, Abhinav Gupta, & Kenneth Marino**
Carnegie Mellon University
{vchen2,abhinavg,kdmarino}@cs.cmu.edu

## ABSTRACT

Complex, multi-task problems have proven to be difficult to solve efficiently in a sparse-reward reinforcement learning setting. In order to be sample efficient, multi-task learning requires reuse and sharing of low-level policies. To facilitate the automatic decomposition of hierarchical tasks, we propose the use of step-by-step human demonstrations in the form of natural language instructions and action trajectories. We introduce a dataset of such demonstrations in a crafting-based grid world. Our model consists of a high-level language generator and low-level policy, conditioned on language. We find that human demonstrations help solve the most complex tasks. We also find that incorporating natural language allows the model to generalize to unseen tasks in a zero-shot setting and to learn quickly from a few demonstrations. Generalization is not only reflected in the actions of the agent, but also in the generated natural language instructions in unseen tasks. Our approach also gives our trained agent interpretable behaviors because it is able to generate a sequence of high-level descriptions of its actions.

## 1 INTRODUCTION

One of the most remarkable aspects of human intelligence is the ability to quickly adapt to new tasks and environments. From a young age, children are able to acquire new skills and solve new tasks through imitation and instruction (Council et al., 2000; Meltzoff, 1988; Hunt, 1965). The key is our ability to use language to learn abstract concepts and then reapply them in new settings. Inspired by this, one of the long term goals in AI is to build agents that can learn to accomplish new tasks and goals in an open-world setting using just a few examples or few instructions from humans. For example, if we had a health-care assistant robot, we might want to teach it how to bring us our favorite drink or make us a meal in just the way we like it, perhaps by showing it how to do this a few times and explaining the steps involved. However, the ability to adapt to new environments and tasks remains a distant dream.

Previous work have considered using language as a high-level representation for RL (Andreas et al., 2017; Jiang et al., 2019). However, these approaches typically use language generated from templates that are hard-coded into the simulators the agents are tested in, allowing the agents to receive virtually unlimited training data to learn language abstractions. But both ideally and practically, instructions are a limited resource. If we want to build agents that can quickly adapt in open-world settings, they need to be able to learn from limited, real instruction data (Luketina et al., 2019). And unlike the clean ontologies generated in these previous approaches, human language is noisy and diverse; there are many ways to say the same thing. Approaches that aim to learn new tasks from humans must be able to use human-generated instructions.

In this work, we take a step towards agents that can learn from limited human instruction and demonstration by collecting a new dataset with natural language annotated tasks and corresponding gameplay. The environment and dataset is designed to directly test multi-task and sub-task learning, as it consists of nearly 50 diverse crafting tasks.[1] Crafts are designed to share similar features and

---

[1]Our dataset, environment, and code can be found at: https://github.com/valeriechen/ask-your-humans.

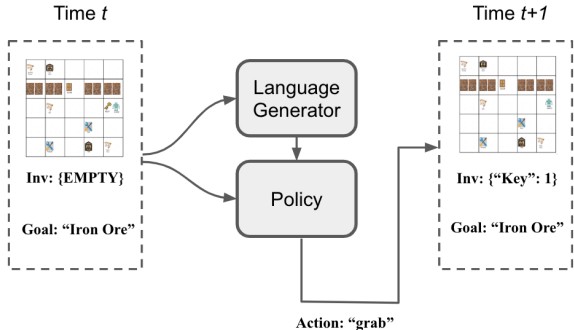

Figure 1: From state observation at time step $t$, the agent generates a natural language instruction "go to key and press grab," which guides the agent to grab the key. After the instruction is fulfilled and the agent grabs the key, the agent generates a new instruction at $t + 1$.

sub-steps so we would be able to test whether the method is able to learn these shared features and reuse existing knowledge to solve new, but related tasks more efficiently. Our dataset is collected in a crafting-based environment and contains over 6,000 game traces on 14 unique crafting tasks which serve as the training set. The other 35 crafting tasks will act as zero-shot tasks. The goal is for an agent to be able to learn one policy that is able to solve both tasks it was trained on as well as a variety of unseen tasks which contain similar sub-tasks as the training tasks.

To do this, we train a neural network system to generate natural language instructions as a high-level representation of the sub-task, and then a policy to achieve the goal condition given these instructions. Figure 1 shows how our agent takes in the given state of the environment and a goal (Iron Ore), generates a language representation of the next instruction, and then uses the policy to select an action conditioned on the language representation - in this case to grab the key. We incorporate both imitation learning (IL) using both the language and human demonstrations and Reinforcement Learning (RL) rewards to train our agent to solve complicated multi-step tasks.

Our approach which learns from human demonstrations and language outperforms or matches baseline methods in the standard RL setting. We demonstrate that language can be used to better generalize to new tasks without reward signals and outperforms baselines on average over 35 zero-shot crafting tasks. Our method uses language as a high-level task to help decompose a larger, complex task into sub-tasks and identify correct sub-tasks to utilize in a zero-shot task setting. We also show that the agent can learn few-shot tasks with only a few additional demos and instructions. Finally, training with human-generated instructions gives us an interpretable explanation of the agent's behavior in cases of success and failure. Generalization is further demonstrated in the agent's ability to explain how the task is decomposed in both train and evaluation settings, in a way that reflects the actual recipes that describe the crafting task. With our dataset collection procedure and language-conditioned method, we demonstrate that using natural human language can be practically applied to solving difficult RL problems and begin solving the generalization problem in RL. We hope that this will inspire future work that incorporates human annotation, specifically language annotation, to solve more difficult and diverse tasks.

## 2    RELATED WORK

Previous works on language descriptions of tasks and sub-tasks have generally relied on what Andreas et al. (2017) calls "sketches." A sketch specifies the necessary sub-tasks for a final task and is manually constructed for every task. The agent then relies on reward signals from the sketches in order to learn these predefined sub-tasks. However, in our setup, we want to infer such "sketches" from a limited number of instructions given by human demonstrations. This setting is not only more difficult but also more realistic for practical applications of RL where we might not have a predefined ontology and simulator, just a limited number of human-generated instructions. In addition, at test time, their true zero-shot task requires the sketch, whereas our method is able to generate the "sketches" in the form of high-level language with no additional training and supervision.

Similarly, other works have used synthetically generated sub-goals and descriptions to train their methods and suffer from similar problems of impracticality. Shu et al. (2018) introduces a Stochastic Temporal Grammar to enable interpretable multi-task RL in the Minecraft environment. Similarly, the BabyAI platform Chevalier-Boisvert et al. (2019a) presents a synthetic language which models commands inside a grid-based environment. They utilize curriculum training to approach learning complex skills and demonstrate through experimentation in their environment that existing approaches of pure IL or pure RL are extremely sample inefficient. Cideron et al. (2019) extend Hindsight Experience Replay (HER) to language goals in the BabyAI platform to solve a single instruction generated from a hand-crafted language. The BabyAI environment is extended by Cao et al. (2020) to include descriptive texts of the environment to improve the generalization of RL agents. Jiang et al. (2019) also uses procedural generated language using the MuJoCo physics engine and the CLEVR engine to learn a hierarchical representation for multi-task RL. Oh et al. (2017) also tackles zero-shot generalizations, but like the others considers only procedurally-generated instructions, learning to use analogies to learn correspondences between similar sub-tasks.

The main work that also investigates using a limited number of human-generated instructions in RL environments is Hu et al. (2019). This paper also uses natural language instructions in hierarchical decision making to play a real-time strategy game involving moving troop units across long time scales. This work uses only behavioral cloning with natural language instructions, whereas we use a mixture of RL and imitation learning. They also do not investigate the benefits of language in zero-shot or few-shot settings and do not demonstrate cross-task generalization as we do.

Hierarchical approaches as a way of learning abstractions is well studied in the Hierarchical Reinforcement Learning (HRL) literature (Dayan & Hinton, 1993; Parr & Russell, 1998; Stolle & Precup, 2002). This is typically done by predefining the low-level policies by hand, by using some proxy reward to learn a diverse set of useful low-level policies (Heess et al., 2016; Florensa et al., 2017; Eysenbach et al., 2018; Hausman et al., 2018; Marino et al., 2019), or more generally learning options (Sutton et al., 1999). Our approach differs in that unlike in options and other frameworks, we generate language as a high-level state which conditions the agent's policy rather than handing control over to low-level policies directly.

Other works have shown the effectiveness of using a combination of reinforcement learning and imitation learning. Le et al. (2018) presents a hybrid hierarchical reinforcement learning and imitation learning algorithm for the game Montezuma's revenge by leveraging IL for the high-level controller and RL for the low-level controller demonstrating the potential for combining IL and RL to achieve the benefits of both algorithms. By learning meta-actions, the agent is able to learn to solve the complex game. However, their meta-actions were also hand specified.

Others have utilized natural language for other tasks, including Williams et al. (2018), Co-Reyes et al. (2018), and Andreas et al. (2018) but not focused on the multi-task learning setting. Matthews et al. (2019) demonstrates the use of word embeddings to inform robotic motor control as evidence of particular promise for exploiting the relationship between language and control. Narasimhan et al. (2018) uses language descriptions of the environment to aid domain transfer. The sub-field of language and vision navigation specifically has investigated how to train agents to navigate to a particular location in an environment given templated or natural language (Chaplot et al., 2018; Anderson et al., 2018; Tellex et al., 2011; Mei et al., 2016; Chen & Mooney, 2011; Yu et al., 2018) or to navigate to a particular location to answer a question Das et al. (2018). Similarly to this work, Nguyen et al. (2019) uses human-generated language to find objects in a simulated environment, Zhong et al. (2020) and Branavan et al. (2012) reads a document (i.e. a players manual) to play a variety of games, and Lynch & Sermanet (2020) trains agents to follow both image and language-based goals. All of these works require the agent to read some text at both train and test time and follow those instructions to achieve some goal. In contrast, at test time, our agent only receives a high-level goal, which is what item to craft. Our agent must take the high-level goal as input and generates its own instructions to solve the task. In other words, our task is both instruction following and instruction generation. Related to instruction generation, some work have explored more generally intrinsic motivation for goal generation (Florensa et al., 2018; Forestier et al., 2017). In our work, however, we learn the goals via the human language instructions.

# 3  HUMAN ANNOTATION COLLECTION

The first step of our approach requires human demonstrations and instructions. To that requirement, we built an interface to collect human-annotated data to guide the learning model.

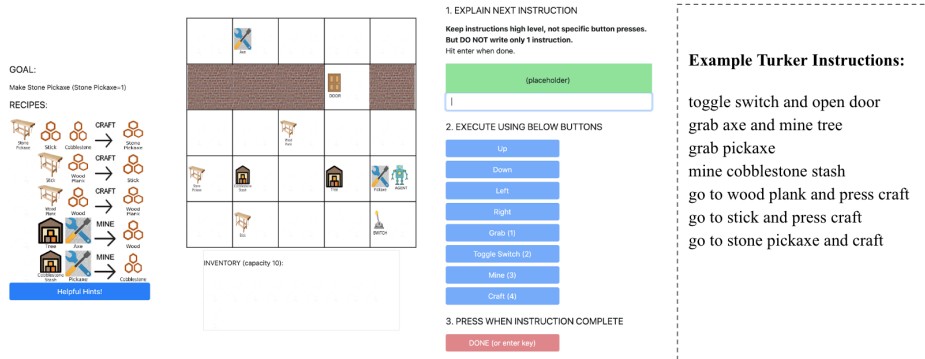

Figure 2: (Left) Example view of game interface that the worker would see on AMT. On the left the worker is given the goal and recipes; the board is in the middle; the worker provides annotations on the right. (Right) Example sequence of instructions provided by the Turker for the given task of Stone Pickaxe.

**Crafting Environment**: As shown in Figure 2, our environment is a Minecraft-inspired 5-by-5 gridworld. The Crafting Agent navigates the grid by moving up, down, left, and right. The agent can grab certain objects, like tools, if it is next to them and use the tools to mine resources. The agent must also use a key or switch to open doors blocking its path. Finally, the agent can also go to a crafting table to build final items. The agent can choose from 8 actions to execute: *up*, *down*, *left*, *right*, *toggle*, *grab*, *mine*, and *craft*. The environment is fully observable. Our crafting environment extends the crafting environment of Andreas et al. (2017) to include obstacles and crafts that are specified by material, introducing compositionally complex tasks (i.e. instead of 'Make Axe', we have 'Make Iron Axe' etc). In total, we consider about 50 crafting tasks, 14 of which we collect annotations for and 35 of which are used for test time. At the start of each game, all object/resource locations are fully randomized in the environment.

**Crafting Task**: The goal of the agent in our world is to complete crafts. By design, a crafting-based world allows for complexity and hierarchy in how the agent interacts with items in the gridworld. To craft an item, the agent must generally first pick up a tool, go to a resource, mine the resource, and then go to a table to craft the item. To make an iron ore, the agent must use the pickaxe at the Iron Ore Vein to mine Iron Ore to complete the task. The Iron Ore recipe is an example of a 1-step task because it creates one item. A 5-step task, like Diamond Pickaxe, involves the mining and/or crafting of 5 items. We capped the tasks at a maximum length of 5 recipe steps to limit the amount of time a worker would have to spend on the task. Note that each recipe step requires multiple time-steps to complete. Crafts are designed to share similar features and sub-steps to test whether the agent is able to learn these shared features and reuse existing knowledge to solve new, but related tasks more efficiently (these relations between tasks are detailed in Table 3 and in Figure 10). While the task may seem simple to human annotators to solve, such compositional tasks still pose difficulties for sparse-reward RL. We further increase the difficulty of this task by restricting the agent to a limited number of steps (100) to complete the task, leaving little room to make unrecoverable mistakes such as spending time collecting or using unnecessary resources.

**Data Collection Process**: Figure 2 shows our interface. Given the goal craft, relevant recipes, and the initial board configuration, the worker provides step-by-step instructions accompanied by execution on the actual game board of each instruction. The workflow would be to type one instruction, execute the instruction, then type the next instruction, and execute until the goal was completed. The data collection interface and a corresponding example set of natural language instructions provided by a Turker is illustrated on the rightmost side of Figure 2. This is but one way that a Turker might choose to break down the 5-step crafting task. The appendix has more details on the collection process in Section A.1. We will release the environment and dataset.

**Dataset Analysis:** Between the 14 crafts, we collected 6,322 games on AMT. In total, this dataset contains 195,405 state-action pairs and 35,901 total instructions. In the supplementary material we present relevant summary statistics about the data, including the number of instructions provided for each $n$-step task. The number of instructions, and consequently, actions required increases with the number steps as shown in Table 4.

## 4 METHODS

Our proposed approach to solving these multi-step crafting tasks is to learn from human-generated natural language instructions and demonstrations. The model is first pre-trained using imitation learning (IL) and then fine-tuned using sparse-reward in reinforcement learning (RL). The goal of the agent is to learn one policy that is able to solve a variety of tasks (around 50) in the environment including ones it has not seen when only trained on a subset of the total tasks.

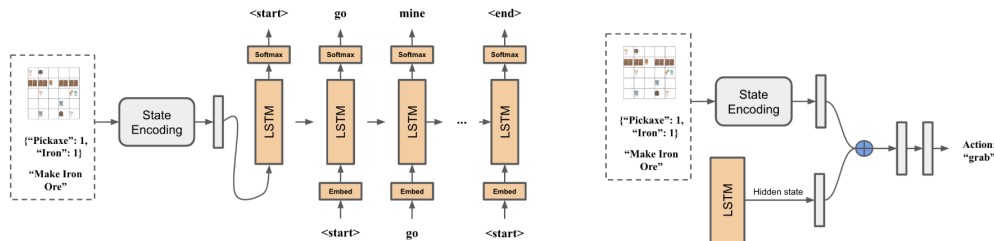

Figure 3: (Left) High-level language generator. (Right) Low-level policy conditioned on language.

**Architecture:** As outlined in Figure 3, we factor the agent into a hierarchical set-up with a language generator at the high-level and policy conditioned on the language at the low-level. At each time step, the state encoder produces a vector representation that is then used as input to both the language generator and language-conditioned policy. Relevant information about the state, including the grid, inventory, and goal are encoded. Items which are relevant for crafting are embedded using a 300-dimension GloVe embedding, summing the embeddings for multiple word items (i.e. Iron Ore Vein). Non-crafting items, such as door, wall, or key, are represented using a one-hot vector. Further details are provided in Section B.

**Imitation Learning Pre-training**: We warm-start the model using the human demonstrations. Language is generated at the high-level with an encoder-decoder framework. The encoding from the state encoder is decoded by an LSTM which generates a natural language instruction. The target language instruction is the AMT worker's provided instruction. In our dataset, the vocabulary size was 212, after filtering for words that appeared at least 5 times. At test time, we do not have access to the ground truth instructions, so instead the LSTM decoder feeds back the previously generated word as the next input and terminates when the stop token is generated. From the language generator module, we extract the last hidden state of the generated instruction. The hidden state is concatenated with the encoded state and passed through a series of fully connected layers. The final layer outputs the action. In the supervised training phase, the full model is trained by backpropagating through a language and action loss (cross entropy loss).

**Reinforcement Learning Fine-tuning**: We use the proximal policy optimization (PPO) algorithm Schulman et al. (2017) in RL with the reward defined below to learn an optimal policy to map from state encoding to output action. The maximum number of steps in an episode is set to 100. We utilize a training set-up which samples from all tasks (1-step through 5-step tasks). In preliminary experiments, we observe that sampling from 3-step tasks alone, for example, poses too complex of an exploration problem for the model to receive any reward. We define a sparse reward, where the agent only receives a reward when it has completed the full craft. In RL fine-tuning, we freeze the language generator component because there is no more language supervision provided in the simulated environment. We also find that empirically backpropagating the loss through language distorts the output language as there is no constraint for it to continue to be similar to human language. All training hyperparameters and details are provided in supplementary materials.

## 5 EXPERIMENTS

We compare our method against five baselines (1-5) which are reduced forms of our method to evaluate the necessity of each component. We also consider two baselines (6-7), which swap out the language generator for alternative high-level tasks, to evaluate the usefulness of language as a selected high-level task. These baselines have the additional training that our method received, as well as the implicit compositionality, but without language. In both baselines (6-7), we perform the same training steps as with our method. Implementation details are presented in Section B.

**1. IL**: The IL baseline uses the same low-level architecture as our method, without a high-level hidden state. The model learns to map state encoding to an output action.

**2. IL w/ Generative Language**: IL w/ Generative Language is the supervised baseline of our method, which does not include RL reward. This baseline allows us to observe and compare the benefit of having a reward to train in simulation when the model has access to both actions and language instructions.

**3. IL w/ Discriminative Language**: We compare our method to a closely adapted version of the method proposed in Hu et al. (2019) which similarly uses language in the high-level. Rather than generate language, their high-level language is selected from a set of instructions from the collected user annotation. We discuss this adaptation in the Appendix. They consider instruction sets of sizes $N = \{50, 250, 500\}$ and find the best performance on the largest instruction set $N = 500$ which is the size we use in our implementation.

**4. RL**: Another baseline we consider is the reinforcement learning (RL) setting where the agent is provided no demonstrations but has access to sparse-reward in RL. The architecture we use here is the same as the IL architecture. This baseline demonstrates the capacity to learn the crafting tasks without any human demonstrations and allows us to see whether human demonstrations are useful.

**5. IL + RL**: We also consider a baseline that does not incorporate language which is IL+RL. In IL+RL, we pretrain the same IL architecture using the human demonstrations as a warm-start to RL. It is important to note that this baseline does not include the natural language instructions as a part of training. We extract all of the state-action pairs at train a supervised model on the data as in the IL model and then we utilize the RL sparse-reward to fine-tune.

**6. State Reconstruction (SR)**: We train an autoencoder to perform state reconstruction. The autoencoder reconstructs the state encoding and the vector at the bottleneck of the autoencoder is used as the hidden layer for the policy. SR as a baseline allows us to consider latent representations in the state encoding as a signal for the policy.

**7. State Prediction (SP)**: We train a recurrent network, with the same architecture as our language generator, to perform state prediction. The model stores the past 3 states from time $T$ to predict the $T + 1$ state. So at time $T$, the states $T - 2$, $T - 1$, and $T$ are used to predict state $T + 1$. From the LSTM, the hidden state is extracted in the same manner as our IL + RL w/ Lang model. SP as a baseline allows us to compare against another recurrent high-level method with the additional computation power.

### 5.1 RESULTS

**Standard setting**: We evaluate the various methods on crafts which we have collected human demonstrations for to benchmark comparative performance in our environment. An initial analysis is to first consider how much the IL model is able to learn from human demonstrations alone, so we consider IL, IL with Generative Language, and IL with Discriminative Language (results are in Section C). None of these approaches are able to solve the most difficult 5-step tasks or the simpler tasks consistently, with an average of about 18-19% success rate for 1-step tasks. We believe the 3 and 5-step tasks are difficult enough such that annotations alone were not able to capture the diversity of board configurations for the variety of crafts given that the board is randomly initialized each time. However, based on an analysis of the language selected (see Tables 11 vs. Table 12), the generated language is more interpretable and made more sense in a zero shot setting. Given the language is fixed after this point, all remaining experiments moving forward use generative language.

As shown in Figure 4, our method performs well against baselines. We find that human demonstrations are necessary to guide learning because the learned behavior for RL is essential to arbitrarily

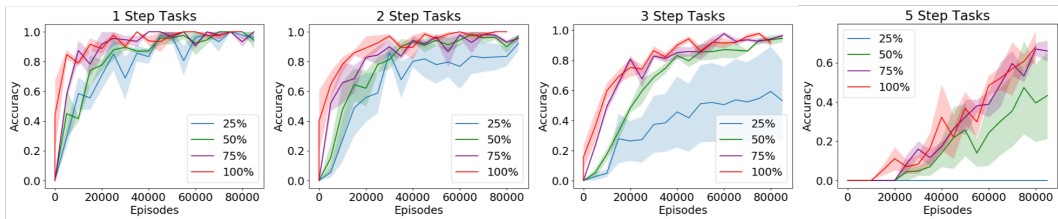

Figure 4: Comparing baselines with our method on accuracy. Human demonstrations are necessary to complete tasks with 3 or more steps. Averaged over 3 runs.

walk around the grid and interact with items. For simple 1 and 2 step tasks, this is a feasible strategy for the allotted steps for an episode. However, there is little room for error in the most difficult 5-step tasks, as even human demonstrations take on average 40 steps to solve. We also find that for the standard setting, incorporating a high-level network allows the model to achieve good results when comparing our method to SP and SR.

In Figure 5 we show the result of our method when we ablate the number of demonstrations we use. This lets us see how many demonstrations we would feasibly need for the model to learn how to solve the crafting tasks. As we decrease the amount of data provided, we find that there is greater variance in the policy's ability to complete the task, but the performance only significantly degrades when we start using only 25% of the data on the hardest tasks.

Figure 5: Ablation of our method with varying amounts of human annotations (25%, 50%, 75% and 100%). For each fraction, we sample that number of demonstrations from the dataset for each type of task. Averaged over 3 runs.

**Zero Shot**: Our method is able to use natural language instructions to improve performance on difficult tasks in the standard setting. But how well is our method able to do on completely new tasks not seen during training? We investigate our performance on zero-shot tasks, where the agent receives no human demonstrations or instructions, and no rewards on these tasks. The agent has to try to complete these tasks that it has never seen before and cannot train on at all. These unseen tasks do share sub-task structure with tasks which were seen in the training process, so the desired behavior is for the model to reuse subpolicies seen in other contexts for this new context. For example, in training the agent might have seen demonstrations or received rewards for a task like "Cobblestone Stairs" and "Iron Ingot." At test time, we can evaluate the agent on an item like "Cobblestone Ingot", which has never been seen by the agent. The agent should be able to infer the sub-task breakdown given prior knowledge of similar tasks.

We present 35 examples of unseen tasks in Table 1. We find that overall our method outperforms all other baselines. While SR and SP were able to match our method's performance in standard setting, they are not able to generalize. SR and SP are viable solutions to learn complex tasks in the standard RL setting, but the representations these models learned do not aid in generalizing to unseen tasks. Here, we believe, using language is key because it creates a representation that better abstracts to new tasks. In the supplementary material, we show that in the cases of unseen tasks, the model indeed is able to generate language that properly corresponds to these new combinations of materials and items, particularly decomposing the complex item into sub-tasks that were previously seen in the training phase.

**Demonstration Only and Few-Shot**: In the demonstration only, we assume that we have access to only human demonstrations for some subset of tasks. From the entire pool of 14 tasks we collected demonstrations for, we withhold 3 tasks (around 20% of total tasks) for testing. These 3 tasks consist of a one, two, and three step task. We run results on 3 permutations of withholding 3

Table 1: Accuracy evaluated on 100 games for 35 unseen crafts. Our method outperforms baselines. We do not list IL or IL w/ Language results which are 0% for all tasks.

| Steps | Diamond Stairs | Cobblestone Boots | Stone Boots | Brick Boots | Cobblestone Leggins | Stone Leggins | Brick Leggins | Diamond Leggins | Cobblestone Door | Stone Door | Brick Door | Diamond Door | Cobblestone Chestplate | Stone Chestplate | Brick Chestplate | Diamond Chestplate | Cobblestone Helmet | Stone Helmet | 2-AVG |
|---|---|---|---|---|---|---|---|---|---|---|---|---|---|---|---|---|---|---|---|
| | 2 | 2 | 2 | 2 | 2 | 2 | 2 | 2 | 2 | 2 | 2 | 2 | 2 | 2 | 2 | 2 | 2 | 2 | 2 |
| RL | 93 | 91 | 95 | 90 | 92 | 92 | 91 | 81 | 0 | 0 | 0 | 0 | 0 | 0 | 13 | 0 | 72 | 28 | 46 |
| IL+RL | 92 | 98 | 85 | 94 | 91 | 83 | 95 | 97 | 21 | 96 | 37 | 18 | 97 | 82 | 97 | 93 | 94 | 91 | 81 |
| SP | 0 | 20 | 0 | 33 | 37 | 2 | 69 | 2 | 90 | 0 | 98 | 0 | 89 | 1 | 78 | 1 | 74 | 2 | 33 |
| SR | 96 | 64 | 0 | 0 | 67 | 70 | 60 | 99 | 0 | 79 | 88 | 74 | 69 | 37 | 16 | 98 | 70 | 0 | 55 |
| Ours | **99** | **99** | **100** | **100** | **93** | **99** | **100** | **100** | **97** | **98** | **99** | **99** | **99** | **100** | **97** | **100** | **99** | **97** | **98** |

| Steps | Leather Stairs | Wood Boots | Cobblestone Ingot | Stone Ingot | Gold Ingot | Brick Ingot | Diamond Ingot | Wood Ingot | Wood Leggins | Leather Door | Wood Chestplate | Wood Helmet | 3-AVG | Brick Pickaxe | Leather Pickaxe | Wood Pickaxe | Iron Pickaxe | Gold Pickaxe | 5-AVG | Overall-AVG |
|---|---|---|---|---|---|---|---|---|---|---|---|---|---|---|---|---|---|---|---|---|
| | 3 | 3 | 3 | 3 | 3 | 3 | 3 | 3 | 3 | 3 | 3 | 3 | 3 | 5 | 5 | 5 | 5 | 5 | 5 | – |
| RL | 1 | 0 | 0 | 0 | 0 | 0 | 0 | 0 | 0 | 0 | 0 | 0 | 0 | 0 | 0 | 0 | 0 | 0 | 0 | 23 |
| IL+RL | 90 | 87 | 0 | 0 | 0 | 0 | 0 | 0 | 85 | 29 | 86 | 87 | 39 | 0 | 0 | 0 | 0 | 0 | 0 | 55 |
| SP | 89 | 0 | **12** | 0 | 4 | **10** | 0 | **47** | 0 | 1 | 26 | 0 | 16 | 0 | 0 | 0 | 0 | 0 | 0 | 22 |
| SR | 1 | 0 | 2 | 0 | 12 | 0 | 0 | 0 | 0 | 38 | 6 | 0 | 5 | 0 | 0 | 0 | 0 | 0 | 0 | 30 |
| Ours | **97** | **98** | 2 | **3** | **18** | 0 | **40** | 0 | **96** | **39** | **95** | **98** | **49** | **36** | 0 | 0 | 0 | **14** | **10** | **69** |

tasks. For each of the 3 withheld tasks, we include these demonstrations in the supervised training phase but do not provide reward in RL fine-tuning. We vary the amount of demonstrations that are provided: 5%, 10%, and 100%. The most generous case is to assume that the model has access to all demonstrations that were collected in the dataset. Per task, the total number of demonstrations was about 300-500. Additionally we considered a more strict few-shot case where we reduce the number of demonstrations to 20-40 which is about 5-10% of the original number of demonstrations. We do not include 5-step tasks because we only collected demonstrations for two 5-step tasks. From the results in Table 2, we can see that our method outperforms baselines in its ability to utilize the few demonstrations to improve performance.

Table 2: Evaluation of few-shot tasks for our method against baseline comparisons. We consider three settings for how many demonstrations are given to the model: 5% (20 demos), 10% (40 demos), 100%. Variance results are included in supplementary material. Results are averaged across 3 seeds.

| | IL | | | IL w/ Lang | | | IL+RL | | | SP | | | SR | | | Ours | | |
|---|---|---|---|---|---|---|---|---|---|---|---|---|---|---|---|---|---|---|
| Steps | 5% | 10% | 100% | 5% | 10% | 100% | 5% | 10% | 100% | 5% | 10% | 100% | 5% | 10% | 100% | 5% | 10% | 100% |
| 1-step | 16% | 18% | 18% | 17% | 19% | 19% | 96% | 91% | 98% | 96% | 98% | 97% | 53% | 84% | 94% | 97% | 90% | 95% |
| 2-step | 4% | 3% | 0% | 5% | 5% | 9% | 66% | 64% | 66% | 53% | 64% | 71% | 10% | 40% | 63% | 87% | 73% | 82% |
| 3-step | 1% | 2% | 0% | 1% | 3% | 4% | 1% | 23% | 22% | 10% | 27% | 46% | 0% | 31% | 50% | 5% | 47% | 74% |

**Interpretability**: One key benefit of incorporating natural language into the model is the ability for humans to interpret how the model is making decisions. We observe that the generated instructions closely match those of the recipes that we provide to the annotators in the data collection phase in both train (Table 12) and test (Table 13, 14) settings. However, the discriminative language didn't break down the task into steps that made sense (Table 11). Figure 6 presents example instructions generated by our model.

# 6 CONCLUSION

In this paper, we present a dataset of human demonstrations and natural language instructions to solve hierarchical tasks in a crafting-based world. We also describe a hierarchical model to enable efficient learning from this data through a combined supervised and reinforcement learning approach. In general, we find that leveraging human demonstrations allows the model to drastically outperform RL baselines. Additionally, our results demonstrate that natural language not only allows the model to explain its decisions but it also improves the model's performance on the most difficult crafting tasks and further allows generalization to unseen tasks. We also demonstrate the model's ability to expand its skillset through few additional human demonstrations. While we demonstrate our approach's success in a grid-based crafting environment, we believe that our method is able to be adapted towards generalizable, multi-task learning in a variety of other environments.

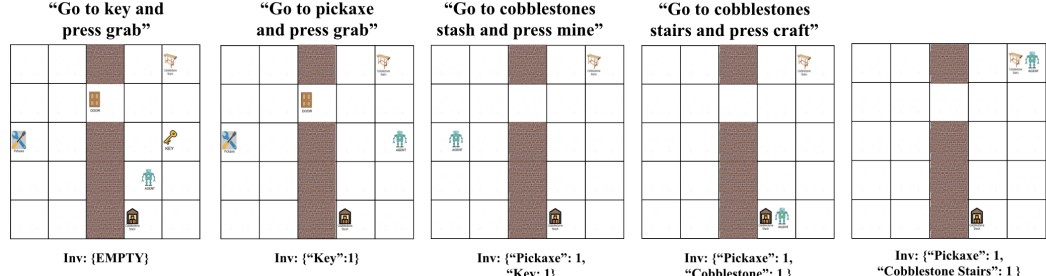

Figure 6: Generated language at test time for a 2-step craft. We only display key frames of the trajectory which led to changes in the language. These key frames match changes in the inventory to the object mentioned in the generated instruction. Qualitatively, the generated instructions are consistent during what we would describe as a sub-task. Quantitatively, the network will spend on average 4.8 steps in the environment for the same generated language output.

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

# A DATASET

## A.1 COLLECTION PROCESS

In this section, we provide additional details for our data collection process on Amazon Mechanical Turk (AMT). Firstly, we filter workers by the following criteria: a) HIT Approval % of greater than 95, b) Location is in an English speaking country, c) soft block has not been granted. These criteria help to ensure the quality of collected data, particularly in terms of natural language.

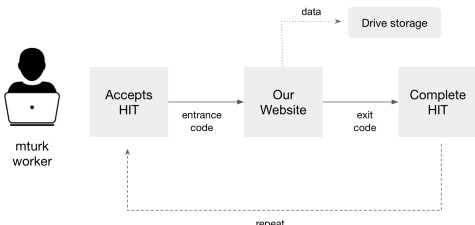

Figure 7: Workflow to collect human demonstrations for our dataset.

We collected the dataset over the course of a few weeks. For each HIT, we paid the Turker $0.65. On average the task took about 3-4 minutes. For each HIT, we generate a unique entrance code that is provided to the Turker on the AMT website. The Turker is also provided with a unique exit code once the HIT is complete. Given the entrance and exit code we are able to pay workers accordingly for their demonstrations.

Workers were provided with an entrance code at the beginning of the task to enter the website and an exit code when they completed the task to be able to submit the HIT. This enforces that we do not have workers doing extra HITs that we are unable to pay for and to ensure that worker who submit HITs have indeed completed our task. Then we also wrote a parsing script to be able to quickly verify all submitted HITs to the task before payment. Specifically, we used this script to manually review the list of instructions generated for each task to ensure that the instructions provided were indeed pertinent to the task at hand.

We had many returning workers who completed our task. A few even wrote emails to our requesting email to let us know that they really enjoyed the HIT and thought it was quite interesting to work on. This suggests that collecting demonstrations of this kind is relatively interesting for humans to provide.

## A.2 HIT INSTRUCTIONS

Prior to starting the task, each worker was provided with this instructions page which gives an analogy for cooking stir-fry as an analogy for the types of instructions that we believe they would provide. The demo page is shown in Figure 8.

Invoking users to provide a particular level of specificity was difficult to convey without explicitly providing example instructions. We deliberately chose not to provide examples as to not prime the worker to a particular format to follow. New workers were given two short games to complete to familiarize themselves with the environment. Returning workers were given one longer game to complete as they already had experience with the task. Workers who completed the task as previously described were fully compensated.

Originally, some workers provided not enough instructions, meaning that they wanted to finish the task as quickly as possible, and other workers provided instructions that were too granular, meaning that they did not abstract the task into sub-tasks and rather wrote "press left" or "go up 1" as their instruction. Prior to approving the HIT, checked prior to the worker submitting the HIT and built in precautions so that they had to redo a level if they did not comply with such instructions clearly delineated in the demo.

### Demo

Welcome to the HIT! **Read instructions carefully. If you do not follow the instructions below, we reserve the right to not pay for the HIT.**

Below is a snippet of a demo from Wikihow of the type of annotation we will be looking for, but in our game setting.
These are their steps to make vegetable stir-fry.

Notice how the tutorial gives you a one sentence description of each high level step, but it is not too specific about each action.

**1. Select vegetables to use.**

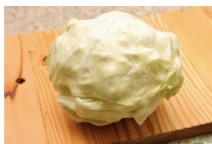

**2. Wash and dry the vegetables..**

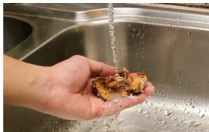

**3. Slice the vegetables into thin pieces.**

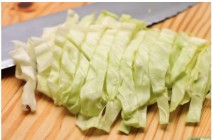

Credits to https://m.wikihow.com/Stir-Fry-Vegetables.

Similarly, in our task, you will be given a goal for the agent to accomplish.

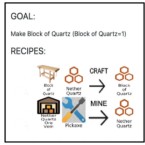

And we want you to write the instruction steps to achieve this goal.

1. Keep it high level and break the task down, just like Wikihow.
2. Do not write just the specific action (e.g. up, down, left, craft, mine, etc) or something like "press up" or "press left 2 times" which are not meaningful steps as your instruction.
3. Since we want high level steps, this means you will execute more than 1 action per instruction.

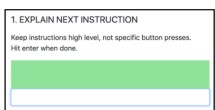

Then, demonstrate how to do this step by executing the step using the UP, LEFT, DOWN, RIGHT (or arrow keys).
Use the other buttons (or keys denoted in the parentheses), explained below to complete the task.

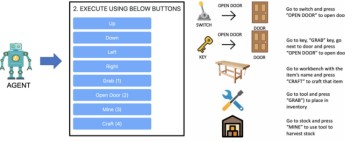

After you have completed the first instruction, press the "DONE" button (or hit enter).

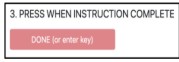

Then return to write the next instruction, then execute, and do this step-by-step until goal is completed.

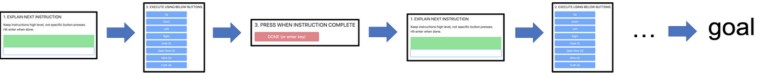

**By the end of this game, you should have typed MULTIPLE instructions, each followed by a sequence of button/key presses.**

**You should be writing meaningful high level instructions which are not "up", "press up", or "go left 2 times".**

**If you do not follow these instructions you will not be paid.**

Figure 8: Demo instructions for AMT workers.

### A.3 ADDITIONAL ENVIRONMENT AND CRAFTING TASK DETAILS

Figure 9 gives an example of the type of task (Make Iron Ore) that would be presented to the worker, which includes the goal, recipes, and the current board. Table 3 shows how the tasks are related in terms of sub-tasks. This might be because of a similar material (i.e. Iron) or a similar craft (i.e. Stairs).

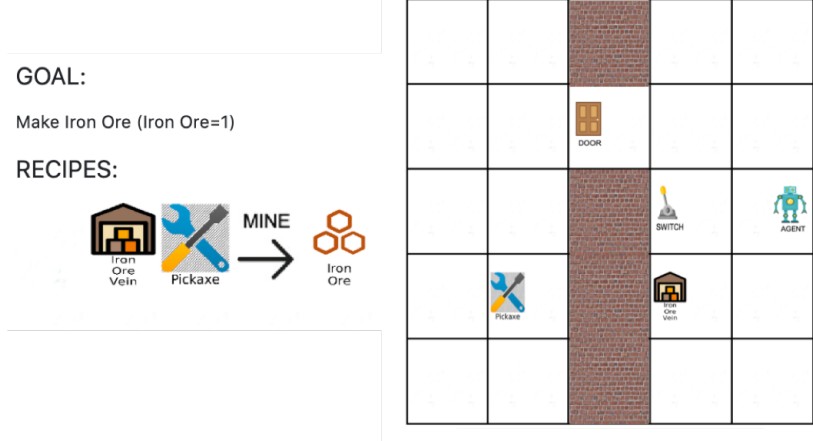

Figure 9: Example board and goal configuration where the goal is to make an iron ore. The worker uses the recipes provided to give appropriate instructions and execute accordingly.

Table 3: List of recipes for which we have collected annotations, labeled by the number of steps needed to complete it and other recipes which may share sub-tasks of underlying structure.

| ID | Recipe Name | Steps | Related Crafts by ID |
|----|-------------|-------|----------------------|
| 1 | Gold Ore | 1 | 2 |
| 2 | Iron Ore | 1 | 1, 8 |
| 3 | Diamond Boots | 2 | 12, 14 |
| 4 | Brick Stairs | 2 | 5, 7 |
| 5 | Cobblestone Stairs | 2 | 4, 7, 13 |
| 6 | Wooden Door | 3 | 7 |
| 7 | Wood Stairs | 3 | 4, 5, 6 |
| 8 | Iron Ingot | 3 | 2 |
| 9 | Leather Leggins | 3 | 10, 11, 12 |
| 10 | Leather Chestplate | 3 | 9, 11, 12 |
| 11 | Leather Helmet | 3 | 9, 10, 12 |
| 12 | Leather Boots | 3 | 3, 9, 10, 11 |
| 13 | Stone Pickaxe | 5 | 5, 14 |
| 14 | Diamond Pickaxe | 5 | 3, 13 |

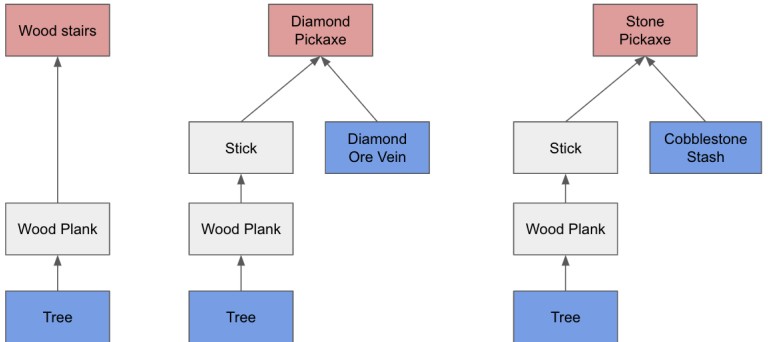

Figure 10: A more in-depth example of 3 out of the 14 training tasks to show how the subtasks are related (red boxes = final craft, blue boxes = raw material).

Table 4: Summary statistics for tasks of varying difficulty.

| Steps | Average # of Instructions | Average # of Actions |
|-------|---------------------------|----------------------|
| 1-step | 3.7 | 15.4 |
| 2-step | 4.9 | 21.5 |
| 3-step | 6.1 | 27.6 |
| 5-step | 8.8 | 40.1 |

## A.4 EXAMPLE DATA COLLECTED

Table 5 gives examples of instructions randomly sampled from our dataset. Even with a limited number of crafts, we were able to collect language instructions with diversity in sentence construction. There are inconsistencies in capitalization and spelling which we handled in the preprocessing of the data. Table 6 shows the most frequently used instructions. Figure 11 gives summary statistics of the instruction side of the dataset.

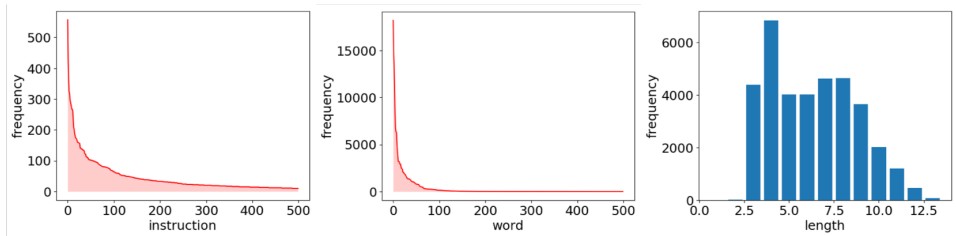

Figure 11: (Left) Instruction frequency (Middle) Word frequency (Right) Histogram of instruction lengths.

Table 5: Examples of randomly sampled instructions.

| | |
|---|---|
| Grab the pickaxe. | Craft diamond axe with its bench, stick, diamond. |
| Make a stone pickaxe from stick and cobblestone. | Pick up key using "GRAB". |
| Go to leather boots bench and Craft Leather Boots. | Grab the key; this may be useful. |
| Move to Tree. | Collect Tool and Pass Through Door. |
| Craft at the leather helmet bench | Go to brick stairs bench and craft. |
| Make a leather chestplate from the leather. | Move up to the axe and pick it up. |
| Unlock the door. | Use Mine on Iron Ore Vein to collect Iron. |
| Move up two squares and grab the key. | Use pickaxe to mine diamond ore vein. |
| Go to swtich and open door with Toggle Switch. | Move to Stick bench and Craft Stick. |
| Chop down the tree to get its wood. | Craft Wood Plank. |
| Go to the stone pickaxe crafting bench. | Harvest wood from tree using "MINE". |
| Mine diamond ore. | 2. "Open Door", then move to Pickax and "grab". |
| Go to stock and click Mine to harvest Cobblestone. | Craft brick stairs with its bench and brick. |
| Toggle the switch to open the door. | Go "Mine" both the iron ore vein and the coal vein. |
| Grab the pickaxe. | Mine the wood. |
| Go back through door and mine gold ore vein. | Mine Diamond Ore Vein. |
| Walk to brick factory and acquire bricks. | Chop down the tree to get its wood. |
| Go to wood bench and Craft Wood Plank. | Move to Rabbit. |
| Craft Diamond Pickaxe. | Go to tools and click Grab to take each one. |
| Move to ingot. | Grab key. |
| Next step, use axe on tree to get wood. | Go to stick workbench and press craft. |
| Go to iron ore vein and mine for iron ore. | Go to TOOL (Pickaxe). |
| Unlock the door with the key and enter the room. | Go to the stone pickaxe table. |
| Pick up axe and put into inventory. | Move to Wood Plank bench and Craft. |
| Craft at the stick bench. | Mine cobblestone stash, then go to tree. |
| Open the door to your right. | Craft Wooden Door. |
| Mine the cobblestone stash. | Craft diamond boots. |
| Get eht pickaxe. | Use pickaxe to mine iron ore vein. |
| Go to tools and click Grab to take each one. | Flip the switch. |

## B  METHODS DETAILS

Given the dataset of human demonstrations, we convert traces of each game into state-action pairs for training. In the subsequent sections are additional details of training parameters for both the supervised IL training and RL fine-tuning. The computing infrastructure for each experiment was on 1 GeForce GTX 1080 Ti GPU. Each experiment took between a few hours to a few days to run.

### B.1  DATA PREPROCESSING

From the dataset, which had 6,322 game traces, we extracted 195,405 state-action pairs and 35,901 total instructions. This is done by matching an action to the corresponding state within a trace as well as the high-level natural language instruction. Each instruction was edited using a spell checker package to reduce the size of the final vocabulary. In the link above, we provide the cleaned version of the dataset.

### B.2  IL PARAMETERS

Both language generation and language-conditioned policy networks use Cross Entropy Loss and Adam optimizer (learning rate 0.001). In addition, the language loss also includes the addition of doubly stochastic regularization, based on the attention mechanism, and clipping of the gradient norm to a max norm of 3. We train for 15-20 epochs. The batch size used during supervised training was 64. By evaluating after each epoch on 100 randomly spawned games, we find that performance plateaus after that number of epochs. As in the RL reward, we only consider a game to be complete if the final craft is completed within the given number of steps. We use the entire dataset for training, since validation/testing were performed on randomly generated new games.

Table 6: Instruction sorted by usage frequency.

| | |
|---|---|
| Grab pickaxe. | Mine cobblestones stash. |
| Grab the pickaxe. | Pick up the pickaxe. |
| Open the door. | Go to switch. |
| Open door. | Go to pickaxe and click grab to take it. |
| Grab axe. | Go to wood plank and press craft. |
| Craft wood plank. | Go to axe. |
| Toggle the switch. | Get the key. |
| Grab the key. | Mine diamond ore vein. |
| Mine tree. | Mine cobblestones. |
| Grab key. | Go to stick and press craft. |
| Craft stick. | Go to pickaxe and select "grab". |
| Toggle switch to open door. | Make planks. |
| Grab the axe. | Use key to open door. |
| Toggle switch. | Craft diamond pickaxe. |
| Go to pickaxe. | Go to pickaxe and press grab. |
| Go to wood bench and craft wood plank. | Craft stone pickaxe. |
| Grab key to open door. | Unlock the door. |
| Get the pickaxe. | Make sticks. |
| Go to tools and click grab to take each one. | Go to stone bench and craft stone pickaxe. |
| Craft leather. | Mine the diamond ore vein. |
| Go to key grab it go to door and open door. | Go to key. |
| Go to switch and open door with toggle switch. | Go to door. |
| Go to tree. | Mine cobblestone stash. |
| Grab the sword. | Grab axe to mine tree. |
| Go to stick bench and craft stick. | Pick up axe using grab. |
| Mine the tree. | Get the axe. |
| Grab sword. | Go to stocks click mine to harvest wood/stone. |
| Pick up pickaxe using grab. | Craft wood plank with its bench and wood. |
| Mine rabbit. | Use the switch to open the door. |

Table 7: We compare to some related datasets/environments (Chevalier-Boisvert et al., 2019b; Jiang et al., 2019; Hu et al., 2019; Anderson et al., 2018). We don't report dataset size for an environment that generates synthetic language. Note that $\sim$ means limited evaluation, they demonstrate unseen evaluation on one setting only). Most notably, our work focuses on developing a method that performs well on unseen tasks. We want to clarify that unseen means tasks/environments which the agent has never received supervised reward for. This is not the same as generating a new configuration of a task that the agent received reward for in training.

| Dataset | Language | Dataset Size | Task | Unseen Tasks |
|---|---|---|---|---|
| BabyAI | Synthetic | – | Navigation/object placement | No |
| HAL/CLEVR | Synthetic | – | Object sorting/arrangement | $\sim$ |
| R2R | Natural | 10,800 Views | Vision+language navigation | Yes |
| MiniRTS | Natural | 5,392 Games | Real-time strategy game | No |
| Ours | Natural | 6,322 Games | Compositional crafting tasks | Yes |

## B.3 RL PARAMETERS

To fine-tune with RL, we first created a gym environment for the Mazebase game, which at reset time will spawn a new Mazebase game in the backend. In the parameters of the environment, we define the maximum episode steps to be 100. The action space of the environment is Discrete space of size 8 (up, down, left, right, toggle switch, grab, craft, mine) and the observation space is a flat vector that concatenates all state observations. For the PPO algorithm, we use a learning rate of 2.5e4, clip parameter of 0.1, value loss coefficient of 0.5, 8 processes, 128 steps, size 4 mini-batch, linear learning rate decay, 0.01 entropy coefficient, and 100000000 environment steps.

### B.4 ARCHITECTURE DETAILS

#### B.4.1 STATE ENCODING

As shown in Figure 12, the relevant information about the state that is encoded includes the 5x5 grid, inventory, and goal. We have two representations of the 5x5 grid: one with items relevant for crafting and another with a one-hot representation of non-crafting-related items, such as a door, wall, or key. All crafting-related items on the board, inventory, and goal are embedded using a 300-dimension GloVe embedding, summing the embeddings for multiple word items (i.e. Iron Ore Vein). The intuition for this distinction is that for generalization, crafting items should be associated in terms of compositionality, whereas non-crafting items are standalone.

To compute the state encoding, we first passed the two grids, inventory, and goal through separate fully connected layers to reduce to the same dimension and concatenated along the vectors. The final size of the state encoding tensor is (27, 128), where 25 are for the grid, 1 for inventory, and 1 for goal.

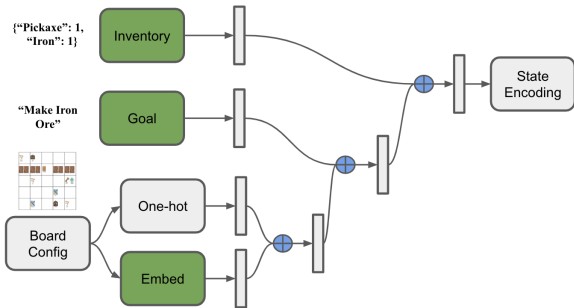

Figure 12: At each time step we encode state relevant observations including the goal, inventory, and grid. This encoding is utilized by both the language generator and the language-conditioned policy. The boxes in green, denote the observations that were encoded using the GloVe embedding.

#### B.4.2 IL W/ DISCRIMINATIVE LANGUAGE

Since Hu et al. (2019) is a closely related work to ours, we wanted to compare our method against theirs. Their method only uses behavioral cloning, which is our IL w/ Generative Language but instead with discriminative language. We modify our high-level language generator to discriminate amongst the most frequent $N = 500$ instructions by adapting code released from Hu et al. (2019) of their LSTM-based language model here). We plugged in our own state encoding instead of theirs, which is anyways tailored to their environment. In summary, the high-level language module is largely the same as our model, both LSTMs, except for the modification of the output layer, which predicts over the set of possible instructions. We similarly extract the hidden state to condition the low-level policy on. The low-level policy is kept the same as our IL w/ Generative Language model for fair comparison. The training parameters are the same as other baselines.

#### B.4.3 OUR METHOD

The state encoder, which is used in the high- and low-level model is covered in the section above. The size of the hidden layer in the language generator LSTM takes input size 128 and has a hidden size of 32. The size of layers in the policy network is 48 and 8, and uses ReLU activations.

#### B.4.4 STATE RECONSTRUCTION

The state reconstruction architecture was instantiated using an autoencoder with 4 hidden layers, taking as input the state encoding tensor. The low-dimensional representation after 2 hidden layers is used as the hidden state for the policy. The autoencoder is trained with MSE loss on the state encoding. This model trained for a total of 25 epochs in the IL phase.

### B.4.5 STATE PREDICTION

The state prediction architecture largely resembled the language generator. However, we removed the GloVe weight embedding layer. In IL training, the dataset was modified to include the past $T$ states. In RL training, the environment was modified to record the previous $T$ states. If there were $< T$ states in the current trajectory, the same state is used as input and subsequently replaced. The recurrent network is trained with MSE loss on the state encoding. This model trained for a total of 20 epochs in the IL phase.

### B.4.6 UNI-MODAL INPUT

A baseline we considered, but did not included in the main text, is to evaluate the necessity of the state encoding in multi-modal datasets. In other works, language instructions are sufficient to solve the task without a state encoding or some representation of the current state. This ablation helps verify that the generated instructions are sufficiently high level that they do not provide the agent with all the information necessary to complete the task in addition to the simulator itself. We considered a baseline where the agent only sees language instructions without the state encoding, so the state encoding is used to generate language, but is not provided as additional input to the policy network. This performs poorly and is not able to solve even the simplest 1-step task. We believe this is because a representation of the current state is critical to completing the task completion, and is not captured by the high-level instructions.

## C SUPPLEMENTARY RESULTS

### C.1 IL RESULTS IN STANDARD SETTING

As shown in Table 8, having access to natural language instructions is only marginally beneficial. While the two environments capture different tasks, we found empirically that the model proposed by Hu et al. [14], which is most similar to our IL+Lang baseline, was able to solve their miniRTS environment using a similarly sized dataset to ours, whereas IL+Lang is not sufficient to complete the most difficult tasks in our environment.

Table 8: Accuracy of IL (with and without language) evaluated over 100 games with 3 different seeds.

| Steps | IL no language | IL Gen. Language | IL Disc. Language |
|---|---|---|---|
| 1-step | 18.00% ± 3.55% | 19.33% ± 1.89% | 20.00% ± 1.35% |
| 2-step | 0.00% ± 0.00% | 9.33% ± 2.05% | 8.33% ± 0.98% |
| 3-step | 0.00% ± 0.00% | 4.33% ± 0.47% | 0.00% ± 0.00% |
| 5-step | 0.00% ± 0.00% | 0.00% ± 0.00% | 0.00% ± 0.00% |

### C.2 DEMONSTRATION ONLY AND FEW-SHOT

As shown in Table 9, we find low deviation in our multiple variance runs. However, we do observe in some cases such as IL+RL 10% and 100% higher variance because in some trials the model was not able to solve any of the 3-step tasks and in others it was. In the cases of low variance, either the model was able to consistently or not solve the tasks.

Table 9: Variance results from Table 3 in the main paper, which presents accuracy.

| Steps | IL | | | IL w/ Lang | | | IL+RL | | | SP | | | SR | | | Ours | | |
|---|---|---|---|---|---|---|---|---|---|---|---|---|---|---|---|---|---|---|
| | 5% | 10% | 100% | 5% | 10% | 100% | 5% | 10% | 100% | 5% | 10% | 100% | 5% | 10% | 100% | 5% | 10% | 100% |
| 1-step | 2% | 1% | 3% | 5% | 3% | 2% | 2% | 5% | 8% | 3% | 1% | 2% | 4% | 3% | 1% | 1% | 2% | 2% |
| 2-step | 1% | 1% | 0% | 1% | 1% | 1% | 9% | 10% | 17% | 19% | 9% | 13% | 8% | 40% | 15% | 1% | 3% | 7% |
| 3-step | 1% | 2% | 0% | 0% | 0% | 0% | 1% | 33% | 31% | 14% | 24% | 18% | 0% | 7% | 15% | 4% | 27% | 10% |

### C.3 REWARD ONLY

Finally, for completeness, we consider the scenario where the agent receives a reward but no demonstrations. The tasks which we select for this setting are sampled from the unseen tasks list. We

choose 3 2-5 step crafts. We evaluate this scenario on our method against other baselines which train using a reward signal. In Table 10, we evaluate on tasks for which we do not have demonstrations and fine-tune a trained model with the reward signal for these tasks. This setting is not very interesting from a generalization perspective, since rewards are a far more expensive resource compared to demonstrations and instructions. We don't include 1-step tasks since that is able to be solved easily by RL alone (see 1-step results in Figure 4). IL and IL w/ Language is not included because this reduces to the zero shot setting.

Table 10: Comparison of 2-5 step tasks where only reward is provided to the agent. We believe IL+RL is not able to adapt to these new tasks, given reward only, since it has overfit to the original training tasks. We find that our method outperforms baselines in this setting.

| Steps | RL | IL+RL | Ours |
|---|---|---|---|
| 2-step | 92.00%±0.81% | 0% | 95.33%±0.94% |
| 3-step | 71.67%±0.47% | 0% | 88.00%±1.41% |
| 5-step | 0.00%±0.00% | 0% | 65.00%±5.67% |

## C.4 INTERPRETABILITY

Table 11: Step-by-step discriminated high-level instructions for seen crafts.

| Goal: Iron Ore
grab the pickaxe
mine iron ore
mine the iron ore vein
unknown | Goal: Gold Ore
unknown
go to the key
unknown
go to gold ore vein and mine |
|---|---|
| Goal: Brick Stairs
grab key and open door
mine bricks
unknown | Goal: Cobblestone Stairs
take the pickaxe
go to cobblestone stash and mine
use pickaxe to mine cobblestone stash
go to cobblestone stash and mine
got to stock and click mine to harvest cobblestones
unknown
craft cobblestone stairs |
| Goal: Diamond Boots
unknown
go to pickaxe
unknown | Goal: Iron Ore
toggle switch to open door
take the pickaxe
toggle switch to open door
unknown |

We present more examples of generated language for both seen and unseen tasks (Table 12 and Table 13). The tables show a complete set of instructions for tasks which were successfully completed. We observe that if the task was not complete then the language generator would be stuck on a particular instruction. The language generated for tasks where the model received supervised data is, as expected, more consistent in using the correct language when generating instructions. However, the language generated for tasks which are new to the model also generated instructions which indicated the use of the correct items. We can observe some mistakes such as generating "leather boots" instead of "leather stairs" or generating an unknown token. Particularly for the Gold/Brick Pickaxe examples, the model is able to generate fairly consistent language for a challenging, new 5-step task. Note that in the Gold Pickaxe example, the model was not able to predict the use of item gold so it uses the word stocks. In the Brick Pickaxe example the model correctly inferred the use of brick. These inconsistencies perhaps can be explained by the amount of training examples where brick and gold appeared, with the former being much more frequent than the latter.

Natural language allows us to understand why a model fails to complete a task from simply looking at the generated instructions, which can facilitate future model development and refinement. We identify two failure cases: (1) failure of language generation and (2) failure of policy. In (1), we

Table 12: Step-by-step generated high-level instructions for seen crafts.

| | |
|---|---|
| Goal: Gold Ore
go to key and press grab.
go to pickaxe and grab.
go to gold ore vein and mine. | Goal: Brick Stairs
go to pickaxe and press grab.
go to the brick factor and mine brick.
go to brick stairs and press craft. |
| Goal: Diamond Pickaxe
go to axe and press grab.
go to key grab it go to door and open door.
go to tools and click grab to take each one.
go to tree and press mine.
go to stocks click mine to harvest.
go to tree and mine.
go to wood plank and press craft.
go to stick bench and craft stick. | Goal: Wooden Door
go to the axe and grab it.
go to the switch and open door.
go to the axe and grab it.
go to the tree.
go to the tree and press mine.
go to wood plank and press craft.
go to wood plank bench and craft wooden door. |
| Goal: Leather Helmet
go to sword and click grab to take it.
go to key and press grab.
go to sword and click grab to take it.
go to rabbit and press mine.
go to leather and press craft.
go to leather boots bench and craft leather. | Goal: Diamond Boots
go to key and press grab.
go to pickaxe and press grab.
go to diamond ore vein and mine.
go to diamond boots and press craft.
go to diamond bench and craft diamond boots. |
| Goal: Iron Ore
go to key and press grab.
go to pickaxe and press grab.
go to iron ore vein and press mine. | Goal: Cobblestone Stairs
go to key and press grab.
go to pickaxe and press grab.
go to cobblestone stash and press mine.
go to cobblestone stairs and press craft. |
| Goal: Wood Stairs
go to axe and press grab.
go to tree and mine.
go to wood plank and press craft.
go to wood stairs and press craft. | Goal: Leather Chestplate
go to sword and press grab.
go to rabbit and mine.
go to leather and craft.
go to leather chestplate and craft. |
| Goal: Leather Leggins
go to sword and click grab to take it.
go to rabbit and press mine.
go to leather and press craft.
go to leather bench and craft leather | Goal: Iron Ingot
go to key and press grab.
go to pickaxe and press grab.
go to iron ore vein and mine.
go to iron ingot and craft. |

observe occasional instructions generated for unseen tasks which have incorrect crafting materials. For example, "Go to iron ingot and press craft" was generated as an instruction for Gold Ingot. In (2), while the generated language is sensical for the task, the policy failed to execute correctly.

Table 13: Step-by-step generated high-level instructions for unseen crafts.

| | |
|---|---|
| Goal: Cobblestone Boots
go to key and press grab.
go to pickaxe and press grab.
go to cobblestone stash and mine.
go to workbench and press craft. | Goal: Diamond Leggins
go to pickaxe and press grab.
go to diamond ore vein and mine.
go to diamond boots and press craft. |
| Goal: Leather Stairs
go to sword and press grab.
go to rabbit and mine the rabbit.
go to leather and press craft. | Goal: Stone Helmet
go to pickaxe and press grab.
go to the cobblestones stash and mine.
go to the workbench and craft. |
| Goal: Diamond Ingot
go to pickaxe and press grab.
go to diamond ore vein.
go to the workbench and craft. | Goal: Brick Door
go to pickaxe and press grab.
go to the brick factory and mine the brick.
go to the brick stairs and craft. |
| Goal: Brick Pickaxe
go to the pickaxe and grab it
go to the axe and press grab.
go to the tree.
go to the tree and mine.
go to the brick factory and mine.
go to the wood plank and craft.
go to the stick bench and craft stick.
go to stick and craft. | Goal: Gold Pickaxe
go to the pickaxe and press grab.
go to the axe and grab it.
go to the tree.
go to stocks and click mine to harvest <unk>.
go to the tree and mine the tree.
go to wood plank and press craft.
go to stick and press craft. |
| Goal: Diamond Stairs
go to key and press grab.
go to pickaxe and press grab.
go to the diamond ore vein and mine.
go to the bench and craft. | Goal: Wood Chestplate
go to key and grab it.
go to axe and grab it.
go to the tree.
go to tree and mine.
go to wood plank and craft. |

Table 14: Example of instruction and inventory side-by-side for 3 unseen tasks. As in Figure 6 from the main paper, the inventory changes when a subtask, given by the instruction, is completed.

| Goal: Leather Door | |
|---|---|
| **Instruction** | **Inventory** |
| go to the sword and grab it | {'Sword': 1} |
| go to the rabbit and mine | {'Sword': 1, 'Rabbit Hide': 1} |
| go to the leather and press craft | {'Sword': 1, 'Rabbit Hide': 0, 'Leather': 1} |
| go to the leather boots bench and craft leather | {'Sword': 1, 'Rabbit Hide': 0, 'Leather': 0, 'Leather Door': 1} |

| Goal: Stone Boots | |
|---|---|
| **Instruction** | **Inventory** |
| go to key and press grab | {'key': 1} |
| go to pickaxe and press grab | {'key': 1, 'Pickaxe': 1} |
| go to the cobblestone stash and mine the <unk> | {'key': 1, 'Pickaxe': 1, 'Cobblestone': 1} |
| go to the bench and craft | {'key': 1, 'Pickaxe': 1, 'Cobblestone': 0, 'Stone Boots': 1} |

| Goal: Diamond Stairs | |
|---|---|
| **Instruction** | **Inventory** |
| go to key and press grab | {'key': 1} |
| go to pickaxe and press grab | {'key': 1, 'Pickaxe': 1} |
| go to the diamond ore vein | {'key': 1, 'Pickaxe': 1} |
| go to diamond ore vein and mine | {'key': 1, 'Pickaxe': 1, 'Diamond': 1} |
| go to the bench and craft | {'key': 1, 'Pickaxe': 1, 'Diamond': 0, 'Diamond Stairs': 1} |

