# OpenReview forum: "Ask Your Humans: Using Human Instructions to Improve Generalization in Reinforcement Learning"
_ICLR.cc/2021/Conference — ICLR 2021 Poster_

### Official Review · AnonReviewer4 · 2020-10-23
**Interesting technique that appears to be surprisingly effective**

**Rating:** 8
**Confidence:** 3

**Review:**

In this paper, the authors present a system that exploits both natural-language instructions, and demonstrations, to learn how to perform multi-subtask tasks in a Minecraft-like environment.  The system has two objectives: First, given a state and objective, learn to generate a natural language description of the high-level subtasks to perform (based on training from human instructions and demonstrations); and second, learn a policy to actually perform the steps.  The policy network accepts both the state description and the final state of the natural language generation network as input at every step.  The system's zero-shot performance is evaluated on previously-unseen tasks with neither demonstrations nor natural-language descriptions provided.

The performance of the network is compared to a number of baselines, some of which are ablated versions of the proposed system, and some of which resemble systems from the literature with similar goals.

The basic idea --- predicting the high-level instructions for a goal, and then using the predicted instructions as an input to the low-level policy --- is a very creative solution to the problems of hierarchical planning, and seems to be effective.  The additional interpretability benefits are very valuable in their own right.  Overall, I think this technique is likely to have a significant impact on future work.

I have some reservations about the clarity of the writing, and the breadth of the empirical comparisons:

The writing is at times difficult to follow, in ways that I think could be straightforwardly addressed, e.g.:
- p.5/6: I found it hard to understand what the difference was between baseline 2 (IL + Generative Language) and baseline 3 (IL + Discriminative Language)
- p.6: "The model stores the past T states as input to predict the T+1 state, where (T=3)." The letter T is probably being incorrectly reused here; maybe it's something like "at time T, the states T-2, T-1, and T are used to predict state T+1", but I can't be sure.

I'm also a little concerned about whether the right baselines were used in the empirical evaluation.  The "ablated" versions of the proposed system are definitely valuable for demonstrating that all of the proposed components are necessary, but I am left wondering how the system's performance compares to the state of the art.  There appears to be only one baseline (#3) that directly corresponds to related work in the literature.  I would appreciate some more direct comparisons to what can already be accomplished in this domain.

---

> ### Author Response · Authors · 2020-11-17
> **Response to Reviewer 4**
>
> Thank you for your comments. First, we address the writing clarifications:
>
> The main difference between generative and discriminative language is whether the instruction is generated token by token or selected from a candidate pool of human instructions. We clarified the description of baseline 3 in the main text and Appendix B.4.2. We additionally clarified baseline 7 in the main text, where your interpretation is indeed correct.
>
> Next, we address your concern about baselines: Our goal is to demonstrate the usefulness of natural language as a high-level controller for generalizing to unseen tasks. As such, baseline 3 is most closely related which uses natural language. Most other methods rely on sketches and templated language, so their approaches aren’t amenable to solving unseen generalization tasks in the same way that ours is. Our environment is also substantially different, thus requiring major adaptations between other work which tackle different problems that we discuss in the Related Work. Any baseline we consider should help us elucidate whether language is useful for generalization, which we believe we have evaluated in depth. As other reviewers have noted, this is a relatively new area. The closest work which we already compare is baseline 3. To work around this lack of comparable work, we additionally proposed baselines 6 and 7 which are stronger than baseline 3. As you mentioned, we believe our work will be an important benchmark for other works to come in the future.

---

### Official Review · AnonReviewer2 · 2020-10-28

**Rating:** 7
**Confidence:** 4

**Review:**

This paper proposes to use natural language to aid reinforcement learning by generating instructions for sub-goals that allow the agent to complete tasks with delayed rewards. The authors first design a multi-task crafting environment and collect step-by-step human demonstrations along with sub-goal instructions using crowdsourced workers. The proposed method then involves a model architecture that includes a recurrent neural network for generating instructions, and a policy network that produces actions conditioned on the state and the instruction. The model is trained using a combination of imitation learning and reinforcement learning. Experiments reveal that the proposed method outperforms several baselines and also generalizes well to zero-shot unseen tasks (where the goal is to craft new combinations of seen components).

Strengths: Overall, the approach is quite novel and interesting, and the experiments (with several baselines) clearly demonstrate the advantage of incorporating language. Paper is clearly written.

Weakness: One aspect that can improve the paper is some more analysis to tease out the effects of using ‘natural’ language vs variations (see points 7, 8 below).

Questions for the authors (M = major questions):
1. (M) From what I understand, the model only uses the current state (grid) as input to generate an instruction at every step. A question I had in my mind while reading was: if so, how does the model know when to switch to the next instruction? I later figured the inventory change is probably the main factor (e.g. after the agent mines some iron ore, it gets added to its inventory), but it would be good to discuss this point a bit in the text since knowing when to switch sub-goals is an important aspect in hierarchical RL.
2. Related to the above point, have you tried providing a few past frames to the input as well?
3. Do you provide the recipes (seen in Figure 2) to the agent during the learning phase? Or is this provided only to the human workers who provide the demonstrations?
4. Thanks for providing details on the data collection in the Appendix. Do you perform any post-processing on the data at all (e.g. have checks to make sure instructions are valid and relevant, correct spelling errors, etc.?)
5. “Items which are relevant for crafting are embedded using a 300-dimension Glove embedding …. Non-crafting items are represented using a one-hot vector.” Why this design choice? Why not represent everything as a GloVe embedding/one-hot vector? I couldn’t find intuition/explanation for this even in the Appendix.
6. “... after filtering for words that appeared at least 5 times.” How do you represent the words that are filtered out in the instruction? Do you use a single out-of-vocabulary (OOV) token for all of them?
7. (M) From the architecture description, the last hidden state of the LSTM is used as input to the policy network, so the generated instruction is not explicitly re-processed by the action generator. While I think this is a neat trick to avoid many issues, the paper leaves open the question of how much of the performance boost is obtained by using ‘natural language’ provided by humans vs just producing a random sequence of tokens. In other words, how much of the performance gain is due to the fact that human language is useful, perhaps because it is (somewhat) compositional? It would be nice to tease out this fact by just generating random instruction sequences for all the train tasks and training the model on them (as a sort of ablation). Note that this is different from the state prediction (SP) baseline which is predicting future states.
8. (M) Related to above, how does an ‘oracle’ baseline which gets access to the human instructions perform on the test tasks? This can provide an (approximate) upper bound for the proposed model.

Missing references:
- Branavan et al., Learning to Win by Reading Manuals in a Monte-Carlo Framework
- Andreas et al., Learning with Latent Language
- Narasimhan et al., Deep Transfer in Reinforcement Learning by Language Grounding
- Schwartz et al., Language is Power: Representing States Using Natural Language in Reinforcement Learning
- Cao et al., BABYAI++: TOWARDS GROUNDED-LANGUAGE LEARNING BEYOND MEMORIZATION
- Luketina et al., A Survey of Reinforcement Learning Informed by Natural Language

---

> ### Author Response · Authors · 2020-11-17
> **Response to Reviewer 2**
>
> Thank you for your detailed comments. We address your questions in order:
>
> 1. Yes, the instruction switch does correspond to changes in the inventory, as seen in Figure 6, which we verify through interpreting the model. The state encoding includes where items are on the board, the goal, and the current inventory. Indeed, the agent associates changes in the current inventory with respect to the goal to its next actions with items on the board. This demonstrates the agent’s ability to plan.
> 2. “Have you tried providing a few past frames to the input as well” This variant is similar to what we consider in baseline 7. Doing this combined approach is, however, a bit tangential to demonstrating the usefulness of language.
> 3. We do not provide recipes to the agent during training or test time. This was designed purely for ease of annotation collection from the human workers.
> 4. We have added some of these details on dataset post-processing to the Appendix in section A.1 and B.1.
> 5. The usage of the GloVE embeddings is motivated by the intuition that we want the model to learn associations between crafting items. For example, if the task is to make an Iron Pickaxe, where similar tasks to construct other Pickaxes have been seen in the training phase then the embedding for the task would reflect this compositionality. However, non-crafting items like the Door is not useful for generalization in this way. We added more information about this intuition in the text in Section B.4.1.
> 6. We use a single out-of-vocabulary token.
> 7. To clarify, is your suggestion to input random words instead of meaningful instructions? It’s not entirely clear to us what this baseline would do, except to ensure that having extra computational parameters from the LSTM makes the model better. If so, some of our current baselines (6 and 7) address this issue, which we believe would provide more meaningful signals than random words (i.e. making it a stronger baseline).
> 8. Test tasks are generated as new boards. As such the human instructions in our dataset would not be able to be used because there would be no annotation that corresponds to the new configuration. To make such an oracle, we would have to have a human in the loop during evaluation to generate an oracle instruction. This might actually lead to lower performance because while the human instructions might be perfectly accurate, there is the potential for it being out-of-distribution for our low-level policy.
>
> The missing references have also been added, we thank the reviewer for these suggestions.

---

### Official Review · AnonReviewer1 · 2020-10-28
**Language as a latent variable to guide RL agents in challenging environments.**

**Rating:** 5
**Confidence:** 4

**Review:**

This paper studies the problem of generating natural language instructions to guide a policy to generalize to new environments where reward function is not given but a small set of demonstrations can be provided. A natural language generation (NLG) LSTM network and an initial policy network is trained from a labeled data collected from human workers. Environment state is encoded into a state encoding and LSTM is initialized with this vector to decode the correct sequence of instruction tokens. Policy network consumes both the state encoding and LSTM hidden state to output the correct action. This model is fine-tuned via reinforcement learning (RL) where the NLG component (assuming only the LSTM) is fixed and only policy network is updated (assuming state encoder is also updated from policy network). During testing, the instruction is not given but sampled from the NLG module. Experimental results show that language improves the performance, especially in zero-shot learning setting, compared to baselines where language is not incorporated or used naively in a discriminator network.

1-) My main concern is that the proposed model is very incremental compared to the previous work by (Hu et. al. 19). Similar to the previous work, the authors claim that the main contribution of their work is to show that language is crucial to improve the generalizability of RL models where there are multiple subtasks and long-horizon problem setting. The only difference is using RL to fine-tune the model and using LSTM hidden state as input to the policy rather then explicitly encoding the generated instructions. Please detail the main contributions compared to the previous work.

2-) There are several confusing sentences as to whether human instructions are given during testing or not. This sentence "If we want to build agents that can quickly adapt in open-world settings, they need to be able to learn from limited, real instruction data." reads as to adapt effectively during testing, it is critical to learn from limited but real instruction data. Similarly, this sentence "We also show that the agent can learn few-shot tasks with only a few additional demos and instructions." also reads as it is important to use few additional demos as well as instructions to learn in few-shot learning during testing. Please clarify the paper if and where you use human instructions during testing.

3-) The authors claim that template based NLG has drawbacks as summarized in 2nd paragraph of the introduction. Given that instructions are not given but generated during testing, it is not clear why having realistic natural language would be preferable to a template based approach. Instructions are only used as latent variables that are fed to the policy to improve the overall performance and to have some interpretability. Please discuss why a template based approach lacks these properties and why also they can not outperform realistic human-based instructions given that they can be sampled indefinitely. Also, I can understand that in general natural language is much more complex than a fixed grammar can generate but examples from Table-11 and Table-12 suggest that a fixed grammar can generate these instructions.

4-) The proposed model has some similarities to multi-task learning where the NLG is trained along with the policy network. It is not clear if this is the major source of the performance improvement or the explicit conditioning of the LSTM hidden state on the policy network. A baseline where the same model is used but hidden state of the LSTM is not fed into the policy would help.


Additional questions and comments.
- Could you compare the results of the proposed model from Table-9 and Figure-5? There is no significant difference that I can see between the two for the "Ours" model which suggests that demonstrations are not useful for improved performance.
- Why is GloVe vectors are removed in state prediction baseline?
- Could you give results with longer training episodes for "5 step tasks" in Figure-5?
- Could you clarify how much the language diverges from human language when gradient is backpropagated via LSTM?
- Please add sublabels to Figure-2.
- In page 5, "an vector representation" --> "a vector representation"
- In page 6, "eesults" --> "results"
- In page 6, "essentially" --> "essential"
- In appendix A.3., "Table ??" is missing the ref.
- In appendix C1, reference [14] should also show the corresponding authors and years.

---

> ### Author Response · Authors · 2020-11-17
> **Response to Reviewer 1**
>
> Thanks for catching the typos, we have corrected those, and for your comments. We address your main comments first:
>
> 1. We want to re-emphasize the point that our paper has contributions in terms of (1) datasets/environments, (2) the task setup with zero-shot/few-shot evaluation and (3) the method. We want to clarify that because their environment is a strategy game, it does not lend itself to generalization (i.e. the strategy that the agent learns isn’t applied to a different game at test time). In this work, we construct a challenging environment (where baseline 3, adapted from Hu et al.'s paper performs poorly on) and collect user annotations for this environment to demonstrate the use of language to solve 35 unseen tasks when the model was only trained on 14 tasks. We added Table 7 to distinguish between some related work. We also discuss why our environment is interesting and difficult in Section 3 of the main text. In our work, we also go further to explore different high-level controllers besides language (baseline 6 and 7) as a way to more rigorously explore the usefulness of language. For the method, there are similarities between our method and (Hu et. al. 19) as we acknowledge in the paper. The main difference is the use of RL to fine-tune the model and the use of an LSTM to encode hidden states. But this is not a small difference! As we show in Figure 4 that these changes have a significant effect on performance. The use of language in RL and IL is an important area of research that is not studied at all in Hu.
> 2. We never use human instructions at test time, in all settings. When a new board configuration is generated, the agent has to complete the craft without supervision. The difference between zero-shot or few-shot is whether the agent has been exposed to the task during training, where in the former setting the task is completely new to the agent at test time and in the latter setting the task has been seen a “few” times. At both test times, the agent generates its own instructions and follows it without any form of supervision.
> 3. We view the benefit of natural language not only from an accuracy perspective, as you suggest, but also in terms of real-world goals. We can clarify our writing to better motivate this in the text. The long-term goal of this line of research is for machines to understand and learn from natural language as humans do - so this entails learning from natural language. To support this, we show in our few-shot experiments that the agent is able to learn from a few additional demonstrations to solve new tasks. In this way, we can learn new tasks directly from humans without needing to hand-design additional templated, synthetic language or sketches, especially when models trained on clean, synthetic data can struggle generalize to noisy, real-world data (see
> https://arxiv.org/pdf/2007.14435.pdf).That’s why having natural language is preferable.
> While the generated instructions tend to be more rigid, the input instructions have much more variety than the input instructions (Table 5). But this is a strength not a weakness. Part of what our method does is learn a “lower variance” instruction specific to the new task and specific environment from the wide variety of human instructions.
> 4. You’re right that it’s important to isolate the effects of language. We believe we have done this through a few baselines, which capture your suggestion “the same model is used but the hidden state of the LSTM is not fed into the policy would help”. The IL+RL baseline (baseline uses the same policy model but does not feed in the hidden state.
>
> Addressing additional questions/comments:
> - We can clarify the settings, which are very different. Table 10 (previously Table 9) evaluates tasks where only reward is provided, which we discuss in more detail in C.3,  while Figure 5 varies the amount of training data provided in the standard training process.
> - GloVe vectors are not removed, they are still used in the state encoding. This is true for all baselines.
> - To keep results consistent, we chose to train for a fixed number of steps for all baselines. We don’t currently have these results, but we can run them.
> - We observed that the language becomes unintelligible. The sequence of generated characters is fully nonsensical. We acknowledge there might be other ways in which the IL and RL phases might be combined and might explore that in the future.

---

### Official Review · AnonReviewer3 · 2020-10-30
**Ask Your Humans**

**Rating:** 7
**Confidence:** 5

**Review:**

This paper studies whether a model can generate its language instruction to solve long and complex sequential tasks. Importantly, ground-truth (natural) language instruction is only provided during a pretraining phase. The contributions of the authors are the following:
 - The extension of the grid-world Minecraft-like environment with a natural language dataset of 6000 instructions over 35 tasks
 - A proof of concept that agent may generate their own language instructions, and such process is beneficial when generalizing to new tasks in the zero-shot / few shot learnings

I provide below several remarks:

### Introduction:
The introduction is quite standard and correctly define the paper goal and organization. On a personal note, I think that the introduction is even a bit too consensual, and it may lack citations to frame it into the literature. For instance, language is still marginal in the RL literature [1], or the concept goal self-generation with language had a very small literature.
 I would reduce the description of what we want, and detail, what we have achieved so far, and the recurrent pitfalls. It is a bit done in the related work section, but it could also be done in the introduction.

[1] Luketina, Jelena, et al. "A survey of reinforcement learning informed by natural language."IJCAI (2019).

### Related work
 - Although the core related works are mostly cited, I may recommend the authors to draw link with the intrinsic motivation literature with self-goal generation (not intrinsic reward) [2-3].
 - I also recommend adding this missing reference [4]. Indeed, the intuition and training procedure are very similar. The current paper uses instruction sequence, while [4] only generates a single instruction and tries not to use human instruction.
 - the difference with Hu et al 2019, a close work, are well-justified
 - why not drawing links with "sketch" and HRL?

[2] Florensa, Carlos, et al. "Automatic goal generation for reinforcement learning agents." ICML. (2018).

[3] Forestier, Sébastien, et al. "Intrinsically motivated goal exploration processes with automatic curriculum learning." arXiv preprint arXiv:1708.02190 (2017).

[4] Cideron, Geoffrey, et al. "Self-educated language agent with hindsight experience replay for instruction following." ADPRL (2020).

### Human annotation collection

 - The data-collection is well justified. I also like the authors' feedback in the Appendix, as it may be useful to other people. Yet, I am missing a small table to compare with other environments or datasets, e.g., MiniRTS, BabyAI, and R2R. (synthetic vs. real, size, can generate new RL scenario, have human dataset, etc.)
 - I am a bit sad that the authors did not create an oracle to generate synthetic instruction on the fly. I acknowledge that it the authors wanted to promote natural language, but it would have able other research analysis (e.g., upper bond by feeding IL+RL with oracle instruction). Besides, it would allow trying different levels of granularity in the instruction. Note that it is not a negative point in the paper, but I think that the paper could be a lot more impactful this way.
 - the analysis of the dataset in the Appendix is neat and appreciated, with multiple examples, and basic statistics
 - Table 3 is not very clear. Would it be possible to turn it into an (evolution) graph?


### Method:
 - The method is pretty simple and standard (but it is well-justified). I found the concept of high-level representation a bit misleading; why not simply refer as language conditioning rather than low/high-level representation. Please also cite the UFVA framework [5]
 - [Key question]: It is unclear when/how the model generates new language instruction? Can you elaborate?
 - I would recommend mentioning that the code is available in this section. (not lost in the Appendix)
 - RL-finetuning: Did you observe some differences in performance/generalization while backpropagating the loss on the language generator? It is a bit frustrating that you mention such experiments without having a small paragraph in the experiments.
 - How important is the use of GloVE in your experiments? This design choice is somehow surprising as the word embedding could have been learned from scratch.

[5] Schaul, Tom, et al. "Universal value function approximators." ICML. (2015).

### Experiments:
 - The baseline 1-5 are sound. Baseline 3 is never correctly described (neither in the main text nor Appendix). Please update the paper accordingly to reproduce it
 - What is the motivation behind baseline 6-7? 6 is an auxiliary loss, and it is quite orthogonal to the approach. I would have replace 7 by a position, instead of the past to have high-level conditioning. Can you explain me why do you think that the past is a relevant high-level signal? Another valid baseline could be simple intrinsic signals (e.g. RIDE [6]), which can be trivially added in light of the current state representation. (Yet, it is also a bit orthogonal)
 - Figure 4.d is quite interesting, and I am curious whether an intrinsic motivation approach would be successful too.
 - Figure 5: What makes the policy worse: lower initial policy or lower language quality? Could you quantify it in some way?
 - Can you analyze further Table1. Saying that your perform better is not fully satisfactory. Why RL collapse in some cases? Why is SP so erratic? What are the successful features that make such a method works when the other one failed? Can you verify it? How does your model handle the cases? Table 1 is very rich, and I would appreciate if the authors analyze it further.
 - I am not sure to have fully understood the few shot learning. Especially, this point "Additionally we considered a more strict few-shot case where we reduce the
number of demonstrations to 20-40, which is about 5-10% of the original number of demonstrations." I thought we already have 5,10,100% of some instructions for pretraining.
 - Interpretability is interesting. I would recommend the authors to have some quantitative results. What is the overlap with human language (Basic BLEU score on held-out data)? What about performing a human evaluation where the human that follow the instruction from the machine and then report the achievement score (100 games would be enough)

[6] Raileanu, Roberta, and Tim Rocktäschel. "RIDE: Rewarding Impact-Driven Exploration for Procedurally-Generated Environments." arXiv preprint arXiv:2002.12292 (2020).

### Conclusion:
Overall, the paper has numerous interests and complement the existing literature nicely.
The paper has still some weaknesses such as,
 - some analysis could have been explored further, e.g., better stating hypothesis beyond the experiments, better complement accuracies with other quantitative
 - Information is dispatched in multiple places, and it takes some time to have a global understanding of the experimental protocol and models. I would recommend to better organize/cluster the information, and add better references from the main text to the Appendix and between sections
 - the environment could be enhanced with a language oracle to extend the spectrum of experiments

On the other side, I believe the paper also has many take-away such as an interesting new dataset, extensive experiments, a valid-research claim with no overstatement, and an open-source code. Therefore, the paper is still marginally above the acceptance threshold.

---

> ### Author Response · Authors · 2020-11-17
> **Response to Reviewer 3**
>
> Thanks for your thorough suggestions. We have added the suggested related work to the updated pdf. We now address your comments by section:
>
> Human annotation collection:
> - We like your idea for adding comparisons between datasets. We added Table 7 to the Appendix which compares the related datasets.
> - While “creating an oracle to generate synthetic instruction” would have been interesting, it is tangential to our research interest in building agents that can generalize using human-generated instruction. To make such an oracle, we would need to have a human in the loop during evaluation to generate an oracle instruction. This might actually lead to lower performance because while the human instructions might be perfectly accurate, there is the potential for it being out-of-distribution for our low-level policy.
> - The goal of Table 3 is to show crafts that share related sub-tasks, so there isn’t exactly a sense of evolution. We added Figure 10 which illustrates how subtasks are related for 3 out of the 14 train crafts.
>
> Method
> - We’re not sure the UFVA framework is related, could you elaborate?
> - The model generates a new instruction at each time step as we illustrate in Figure 1. We find that for a given “sub-task” the instruction remains the same, even though a new instruction is generated at each time step.
> - We observed that backpropagating the loss in the RL phase messed up the language and worsened performance (i.e. a noisy signal from the high-level controller is confusing for the low-level action). Therefore we decided to freeze the language generator instead.
> - The design choice behind using GloVE was that embeddings provide additional context which we hypothesized might be useful for associating crafting objects in the environment. Given that GloVE is used in all of the baselines in the same way (i.e. the baselines all use the same environment encoding architecture), it should not make a difference.
>
> Experiments
> - We have updated Appendix B.4.2 to better describe baseline 3.
> - The goal of 6 and 7 is to try different high-level signals that might be useful for the agent that does not utilize language. This way these baselines have the additional training that our method received, as well as the implicit compositionality, but without language.
> - “What makes the policy worse: lower initial policy or lower language quality? Could you quantify it in some way?” You raise an interesting point to compare the quality of the low vs high level. However, it’s not entirely clear how language quality or initial policy actually translates to gameplay (i.e. we have observed low loss on training but it doesn’t necessarily mean the agent can play the game).
> - “Why RL collapse in some cases?” RL searches for items to collect without planning how the items will be used downstream, causing it to run out of time. “Why is SP so erratic? What are the successful features that make such a method works when the other one failed?” SP and SR perform well on the training tasks in terms of accuracy. However, in contrast, our method which uses language is able to generalize to new tasks without overfitting to the training tasks better than these other baselines.
> - Few-shot learning is different from the ablation of the standard setting. In the few-shot setting, a third of the tasks receive a few number of examples (i.e. 5-10% of total) and NO RL training while the rest receive the standard number of examples/RL training. We then evaluate test accuracy for that third of tasks. The ablation setting is when all tasks receive 25/50/75% of total examples AND RL training. We find that the few-shot tasks which receive 10% of examples can achieve pretty good results that are comparable to 25% example AND training.
> - While interpretability wasn’t the main focus of this work, if you look at Table 11, you can see that the generated instructions largely reconstruct the helpful recipe guides that we provide for the human annotators (example in Figure 9). We could then hypothesize that if we were to provide these instructions to a human follower, they would also be able to solve the task.

---

### Public Comment · ~Andrew_Chester1 · 2020-11-15
**References to Arxiv preprints**

There are numerous references to arXiv preprints even where those papers have been published in traditional venues. For example:
* Co-Reyes et al. 2018 is published in ICLR 2019
* Eysenbach et al. 2018 is published in ICLR 2019
* Florensa et al. 2017 is published in ICLR 2017

This is not an exhaustive list; I have not checked all the references.

I would encourage the authors to review all of their references to preprints and replace them with published versions where appropriate, to give the clearest picture of the canonical versions of these works.

---

> ### Author Response · Authors · 2020-11-17
> **References updated**
>
> Thanks for your suggestion - we have updated the few missing references accordingly.

---

### Decision · Program_Chairs · 2021-01-07
**Final Decision**

**Decision:**

Accept (Poster)

**Comment:**

Although some reviewers still had concerns about the novelty of the proposed method, most of the other concerns have been addressed in a satisfying manner according to reviewers. They globally have a positive opinion about the paper after revision.